# Amphibious microneedles for programmable delivery of biomolecules and microorganisms in living plants

Di Shen [1], Sivamathini Rajappa [1], Yue Zhao[1], Zhihao Pang[1], Yuliang Li[1], Calvin Thenarianto[1], Suppanat Puangpathumanond[1], Cansu Sevencan [1] & Tedrick Thomas Salim Lew [1,2,3] ✉

Efficient cargo delivery is essential for plant trait engineering, yet existing methods are often species-specific and ineffective across diverse habitats. Here, we develop core-shell microneedles for targeted delivery of biomolecular cargoes and active microorganisms into both terrestrial and aquatic plants. The microneedle architecture is rationally engineered to resist water exposure and release cargo upon contact with plant interstitial fluid, enabling controlled delivery into tissues and cells. We demonstrate that these core-shell microneedles can efficiently transport diverse cargoes, from nanoscale biomolecules such as functional nucleic acids, proteins, and plant hormones to microscale bioactive *Agrobacterium*, leading to strong protein expression and enhanced plant growth. Underwater delivery of salt-tolerance genes into submerged freshwater plants further demonstrates the platform's utility for engineering stress resilience in challenging environments. By facilitating the cellular uptake of diverse cargoes into intact plants across different habitats, this amphibious microneedle strategy offers a versatile cargo delivery tool to advance plant biotechnology and environmental applications.

Plants exhibit remarkable biodiversity, thriving in diverse ecological niches from terrestrial environments to submerged aquatic ecosystems. As key players in ecosystems, they serve as vital sources of food, medicines, and biomaterials[1–4], while acting as essential carbon sinks in their respective habitats[5–7]. Optimizing plant growth and engineering improved traits are therefore crucial for addressing sustainability and food security challenges in the changing climate[8–10]. The success of these endeavors relies on the efficient delivery of nutrients and bioactive molecules, such as nucleic acids and proteins, into living plants. However, existing delivery approaches through foliar spraying, biolistic bombardment, or *Agrobacterium*-mediated transformation suffer from low efficiency, poor target specificity, difficulty in controlling release rate, and limited applicability across plant species[11,12]. Moreover, wet plant surfaces from rain and irrigation reduce delivery

efficiency by washing away applied cargoes[13,14]. In aquatic environments, the effective delivery of nutrients or biomolecules is further hindered by rapid dispersion into the surrounding environment, restricting cargo uptake by intended plants and limiting the accessibility of such plants to engineering and growth improvement despite their critical ecological functions[15]. These challenges underscore the need for a universal delivery platform capable of efficiently transporting functional biomolecules into both terrestrial and aquatic plants for advances in agriculture and aquaculture biotechnology.

An effective and robust delivery platform for plant engineering must overcome plant physical barriers while protecting cargoes from degradation or loss to the environment. The plant cell wall, a multilayered and rigid structure, poses a major challenge to many delivery technologies by limiting cargo penetration. Nanomaterials, such as

[1]Department of Chemical and Biomolecular Engineering, National University of Singapore, Singapore, Singapore. [2]Research Centre on Sustainable Urban Farming, National University of Singapore, Singapore, Singapore. [3]NUS Environmental Research Institute, Singapore, Singapore. ✉e-mail: tedrick@nus.edu.sg

carbon nanotubes, gold and silica nanoparticles[16–18], have emerged as promising delivery vehicles for plant biotechnology mainly due to their nanoscale dimensions, which enable them to traverse cell wall pores[18,19]. These nanomaterials are typically introduced using syringe-mediated pressurized infiltration through the stomatal openings. However, applying this technique to aquatic plants or wet leaf surfaces is technically challenging, as the nanomaterial dispersions readily diffuse and disperse into the surrounding water rather than effectively infiltrating plant tissues. Additionally, some aquatic plants lack stomata, further limiting the efficiency of pressure-assisted infiltration[20]. Similarly, *Agrobacterium*-mediated plant transformation, which is largely administered through pressurized infiltration, is ineffective in aquatic plants due to similar constraints imposed by water-rich environments.

Microneedles provide an alternative, minimally-invasive approach for cargo delivery by creating micron-scale pathways into biological tissues. While widely studied in animal models, their application in plants remains largely unexplored[21–23]. Recent studies demonstrated the use of soluble silk microneedles to deliver hormones in *Arabidopsis thaliana* plants, where rapid microneedle dissolution within 10 min led to over 50% cargo release[24]. However, such fast dissolution would result in premature cargo loss before penetrating target tissues in aquatic plants and wet terrestrial plants environments, such as in humid or irrigated environment commonly found in paddy soils, rendering such dissolvable microneedles unsuitable for water-rich conditions. Alternatively, insoluble microneedles prepared from hydrophobic materials enable slower cargo diffusion but require prolonged application, increasing tissue damage and biological stress[25,26]. Moreover, their fabrication typically involves harsh conditions, such as organic solvents, high temperature, or pressure, which could compromise the bioactivity of cargoes and microorganisms[27]. These previous designs suggest that a robust microneedle platform for efficient cargo delivery to wet plant surfaces should exhibit the following properties: (1) minimal cargo release in wet environments before reaching the intended plants, (2) fast cargo release once the platform is inserted into biological tissues to minimize the duration of microneedle application, and (3) biocompatible components inducing negligible stress responses in plants.

In this work, we develop a water-responsive, programmable microneedle platform for precise and efficient bioactive cargo delivery into both terrestrial and underwater plants. These core-shell microneedles feature a rational biomaterial design, incorporating hydrophilic cores to preserve the bioactivity and viability of biomolecular and microorganism cargoes, and fatty acid-based hydrophobic shells acting as a water-resistant, rate-limiting diffusion barrier. This architecture minimizes premature cargo release in external aqueous environments, with a shell thickness that can be tailored to regulate water diffusion and cargo release kinetics. Furthermore, the microneedles integrate a spatially compartmentalized effervescent formulation that produces controlled bursts of $CO_2$ at the core-shell interface, enhancing penetration and cargo delivery into plant tissues. Upon insertion, biofluid gradually permeates the hydrophobic shell, triggering a delayed effervescent reaction that dissolves the core and releases the cargo with precisely controlled timing. This responsive release behavior, adjustable via core and shell formulations, enables effective delivery of nucleic acids, proteins, nanoparticles and live bacteria across both wet terrestrial plant surfaces and submerged aquatic environments. By offering controlled, efficient, and versatile cargo delivery, this amphibious microneedle platform expands the toolbox of species-independent plant delivery systems for agricultural and aquacultural biotechnology.

## Results
### Design and characterization of amphibious core-shell microneedles
Wet surfaces, commonly present on terrestrial plants with waxy cuticles and submerged aquatic plants, pose significant challenges for cargo delivery in living plants, particularly for microneedle-based platforms (Supplementary Fig. 1). To address this limitation, we sought to develop microneedles with a hydrophobic shell that acts as a water diffusion barrier, preventing premature cargo release in water-rich environment, and a hydrophilic core to maintain cargo viability. To accelerate cargo release in plant biofluids and minimize prolonged microneedle application, we incorporated effervescent reaction precursors into the core-shell structure. Unlike conventional effervescent tablets, which mix acid and bicarbonate in the same compartment and risk uncontrolled reactions under wet conditions (Supplementary Fig. 2a), our microneedles compartmentalize these precursors into distinct core and shell regions (Supplementary Fig. 2b). This design delays the reaction until water permeates the shell to dissolve the acid precursor, triggering $CO_2$ generation at the core-shell interface. The resulting gas bubbles dissolve the core, driving rapid cargo release[28,29]. We hypothesized that this approach would enable programmable burst release of cargo specifically within plant tissues, while remaining unaffected by external wet environments.

The water-responsive microneedle patch (MNP) was prepared through a molding strategy (Fig. 1a, and Supplementary Figs. 3–5). Its hydrophilic core was formulated using sodium hyaluronate (HA) and potassium bicarbonate ($KHCO_3$), rationally selected to maintain a neutral pH-buffering environment and preserve cargo bioactivity. The exterior microneedle shell was prepared by immersing the microneedle core in a dichloromethane (DCM) mixture of decanoic acid (C10 fatty acid) and poly-ε-caprolactone (PCL). Finally, the as-synthesized MNP was left to ambient evaporation to completely remove DCM and form the core-shell MNP. Considering regulatory restrictions on DCM, hexafluoroisopropanol (HFIP) could also be employed as an alternative solvent (Supplementary Fig. 6, and Supplementary Note 1). Both core-only MNP (herein denoted as core MNP) and core-shell MNP showed uniform microneedle arrays, with the core-shell MNP exhibiting a rougher surface compared to the core MNP due to shell formation after evaporation (Fig. 1b). Core-shell MNP demonstrated higher compressive forces than core MNP (Fig. 1c), suggesting that the hydrophobic shell not only provided temporary water resistance but also enhanced mechanical strength to enable more effective insertion into plant tissues. Lower C10:PCL ratio facilitated the formation of a more compact shell structure with decreased crystallinity, as shown in X-ray diffraction (XRD) analysis and scanning electron microscope (SEM) (Fig. 1d, e, and Supplementary Fig. 7)[30]. The distinct compartments of core-shell MNP were visualized by confocal laser scanning microscopy (CLSM) analysis (Fig. 1f). Each microneedle has a pyramidal shape with base width of ~100 μm and height of ~200 μm for both core MNP and core-shell MNP. Their significantly shorter height, compared to conventional microneedle designs used for skin penetration in animal studies[31–33], was rationally designed to align with the thin structure of the leaf mesophyll (400-800 μm in thickness)[34,35]. To form a uniform shell structure, a moderate concentration (2.5% C10 and 7.5% PCL) was chosen for subsequent studies (Supplementary Figs. 8, 9, and Supplementary Note 1).

### Core-shell MNP enables programmable, water-responsive cargo delivery
We hypothesized that cargo release from core-shell MNP upon water exposure occurs in two distinct phases (Fig. 2a, and Supplementary Fig. 10). The first phase (Phase I) is a diffusion-limited regime (Supplementary Note 2). Water comes into contact with the hydrophobic shell and forms nano- or micro-pores as the sparingly soluble fatty acid begins to hydrate and PCL chains undergo slight swelling[36]. In the second phase (Phase II), water reaches the core-shell interface and induces an effervescent reaction, which significantly accelerates cargo release. Here, the effervescence reaction occurs between the hydrated fatty acid and $KHCO_3$, producing fatty acid-potassium salt, $H_2O$ and $CO_2$ (Fig. 2b). The $H_2O$ generated further promotes the hydration of

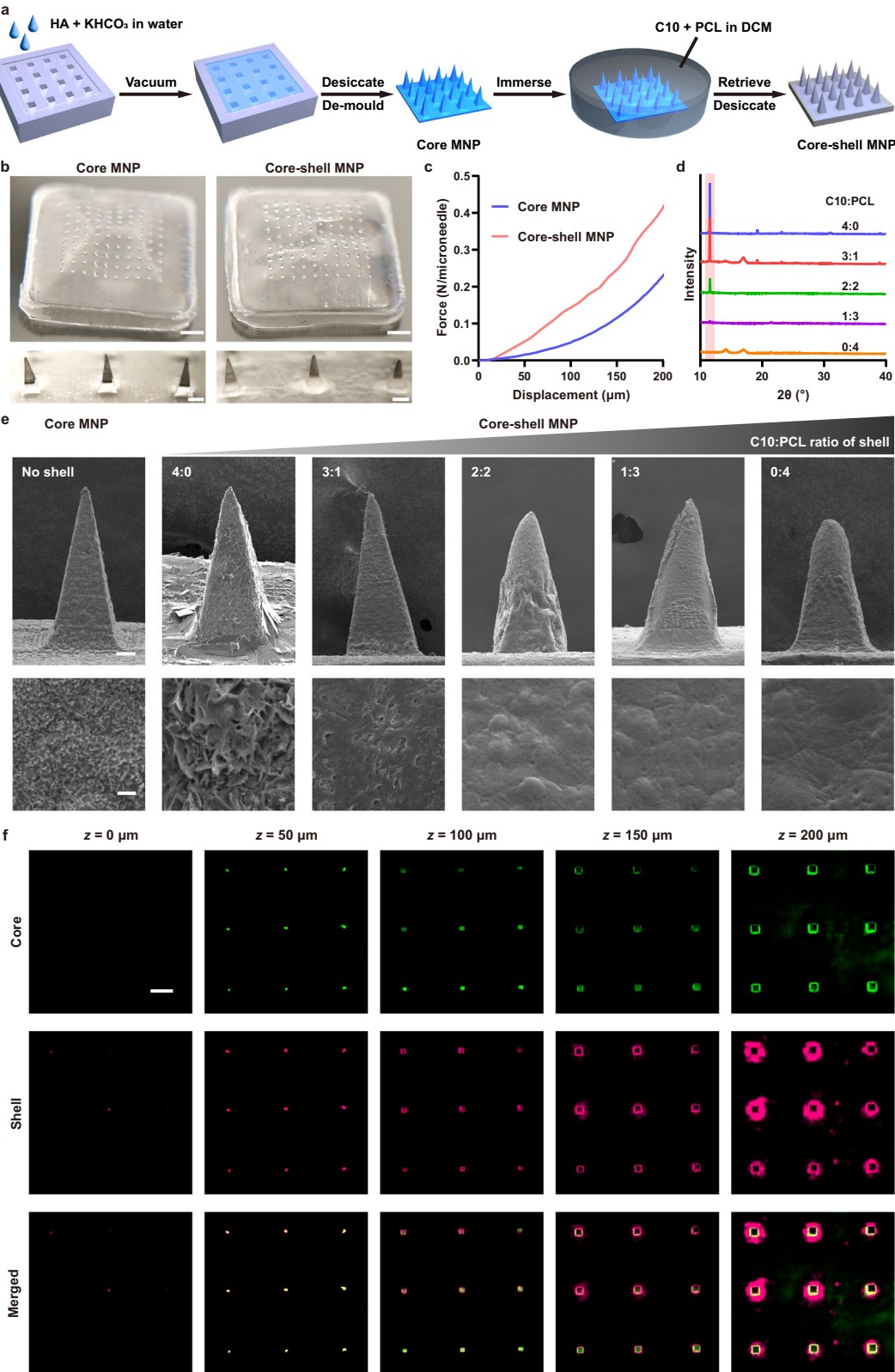

**Fig. 1 | Characterization of the core MNP and core-shell MNP. a** Schematic illustration of the preparation process of the core MNP and core-shell MNP. **b** Optical microscope images of the core MNP and core-shell MNP (scale bar, 1 mm, upper; 100 μm, lower). Imaging was repeated three times independently with similar results. **c** Mechanical strength curves of the core MNP and core-shell MNP. **d** XRD patterns for the shell layer of the core-shell MNP with different C10:PCL

ratios. **e** SEM images of the core MNP and core-shell MNPs (scale bar, 25 μm, upper; 2 μm, lower). Imaging was repeated three times independently with similar results. **f** Representative confocal images showing the respective core and shell components of core-shell MNP at different depth (scale bar, 200 μm). Imaging was repeated three times independently with similar results. Source data are provided as a Source Data file.

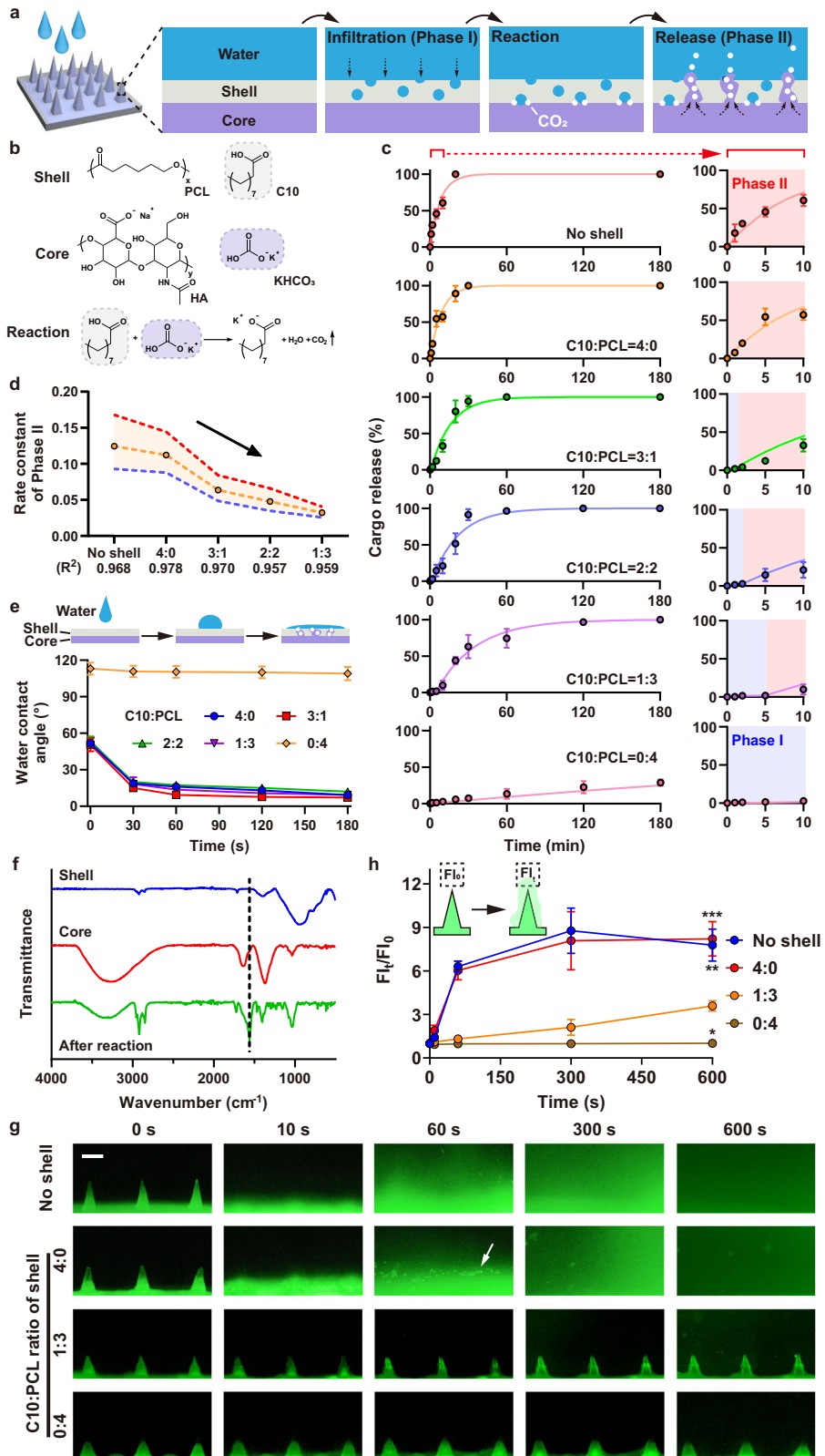

fatty acid and $KHCO_3$, sustaining the effervescent reaction. Simultaneously, bursts of $CO_2$ bubbles create additional pores and cracks in the shell, accelerating water diffusion and cargo release (Supplementary Fig. 11).

To study the release kinetics from MNP in vitro, rhodamine B (RhB) dye was loaded into the MNP core. Immersion of MNP in water is expected to increase the fluorescence signal of RhB as it leaks and diffuses to the external environment. Core MNP showed a rapid cargo-release profile in aqueous environments due to its high hydrophilicity (Fig. 2c, and Supplementary Fig. 12a, b, Supplementary Fig. 13). In comparison, the core-shell MNPs exhibited delayed-release effects, with the duration of Phase I progressively increasing as the C10:PCL

**Fig. 2 | In vitro cargo release from core MNP and core-shell MNP. a** Schematic illustration of the cargo-release mechanism from core-shell MNP. **b** Components of the core and shell regions in the core-shell MNP. The effervescent reaction occurs between C10 (from shell) and $KHCO_3$ (from core). **c** In vitro release profiles of the core MNP (no shell) and core-shell MNPs with different ratios of C10:PCL. **d** Rate constants, $k$, fitted according to the first-order release kinetics. The upper and lower lines represent the upper and lower limits of the 95% confidence interval, respectively. The data points in the middle represented the best fitted values of the rate constant. **e** Water contact angles for the double-layer samples mimicking core-shell MNP. **f** FTIR spectra of the shell, core and the core-shell regions after effervescent reaction. **g** Fluorescence images of core MNP and core-shell MNPs after water contact (scale bar, 200 μm). White arrows denote $CO_2$ bubbles generated from the effervescent reaction. RhB was used as a representative cargo and its fluorescence was depicted in green. Imaging was repeated three times independently with similar results. **h** Time-dependent change of relative fluorescence intensity (FI) of different MNPs upon water contact. $FI_0$ and $FI_t$ represented the total fluorescence intensity (in a 200 μm × 200 μm area) at 0 second and $t$ second after water contact. The selected area was located 100–300 μm above each microneedle. Data of (**c**), (**e**) and (**h**) represent mean ± s.d. ($n = 3$ independent samples). Statistical analysis was performed by one-way ANOVA with Tukey's multiple comparisons test; *$P < 0.05$, **$P < 0.01$ and ***$P < 0.001$ compared with the "1:3" group. Source data are provided as a Source Data file.

ratio decreased (Fig. 2c). Such release behavior could be observed on either hydrophilic or hydrophobic cargoes (Supplementary Fig. 14, Supplementary Table 1). For example, RhB release from core-shell MNP decreased from 54.6% (C10:PCL = 4:0) to 2.1% (C10:PCL = 1:3) within the first 5 min; meanwhile, the time required to release 50% of the loaded cargo (denoted as $t_{1/2}$) increased from 6.2 to 27.0 min (Supplementary Fig. 15). The release profiles in Phase II were also well-described by first-order reaction kinetics, with lower fitted rate constant (k) as the C10/PCL ratio was gradually reduced (Fig. 2d, and Supplementary Fig. 16, Supplementary Note 2).

Furthermore, by employing fatty acid with low solubility such as C14, thickening the shell, or increasing the $HA/KHCO_3$ ratio, the cargo release rate could be systematically reduced, demonstrating the programmability of core-shell MNPs to precisely control cargo release kinetics in aqueous environments (Supplementary Fig. 17). Cargo release in plants was also simulated by immersing MNP in apoplastic fluid extracted from plants. No significant difference in cargo release profile was observed between MNP immersed in water and plant apoplastic fluid (Supplementary Fig. 17j–l). The spatiotemporal dynamics of cargo release were also successfully described by a mathematical model derived using the implicit finite-difference method (Supplementary Fig. 18, and Supplementary Movie 1, Supplementary Note 3). The formation of fatty acid-potassium salt (surfactant) was confirmed by FTIR and contact angle measurements (Fig. 2e, f, and Supplementary Figs. 19 and 20, Supplementary Note 4). The formation of $CO_2$ bubbles and subsequent cargo release were also visualized upon immersing MNP in water (Fig. 2g, h). These observations are consistent with the release kinetic parameters determined by bulk RhB fluorescence measurements, further confirming the important role of effervescence reaction in accelerating cargo release from the MNP in aqueous environment.

To further evaluate cargo release kinetics, core-shell MNPs were inserted into leaf tissues (Supplementary Fig. 12). Minimal cargo release was observed within the first 5 min, consistent with the release profile from water immersion (Supplementary Fig. 12b–d). -88% of the encapsulated cargo was released into leaf tissues over 12 h, with the remaining 12% likely retained in the backing region. To reduce this loss and confirm tip-specific release, cargo was selectively loaded onto the microneedle tips only (Supplementary Fig. 12e, and Supplementary Fig. 21). In this configuration, only 0.6% of the cargo was released within the first 2 min, confirming the delayed onset of release during initial insertion. Full cargo release occurred within 2 h, closely matching the profile observed in water immersion (Supplementary Fig. 12f). These comparisons collectively indicate that cargo release is predominantly mediated through the microneedle tips instead of the backing, and the consistent delayed-release profile across all scenarios supports the system's suitability for precise and controlled delivery into tissues of both terrestrial and underwater plants. Overall, these characterization studies showcase the water-responsive cargo release mechanism in the core-shell MNP and demonstrate the programmability of the design to precisely tune cargo release kinetics in water-rich environments.

## Core-shell MNPs facilitate multiscale cargo delivery in terrestrial and aquatic plants

Existing biomolecule delivery platforms to living plants require pressurized infiltration or biolistics, precluding efficient delivery through wet plant surfaces in terrestrial and aquatic plants. There is also a size restriction on the type of cargoes each delivery technology can deliver. Here, we investigate whether the core-shell MNP can serve as a plant species-independent delivery platform for multiscale cargo delivery. We evaluated the efficiency of core-shell effervescent MNP to deliver cargoes to both terrestrial plants (*Nicotiana benthamiana*, *Brassica chinensis* and *Hordeum vulgare*) and aquatic plants (*Microsorum pteropus* and *Ipomoea aquatica*). These plants were chosen for their importance in plant biology studies and relevance to agriculture-aquaculture nexus as food crops or feedstocks for marine animals. The core-shell MNP could be readily applied to either dry or wet leaves of *N. benthamiana*, with the formation of micropores confirming successful microneedle insertion through the wet surface (Fig. 3a, b, and Supplementary Fig. 22). Similar trend was observed on *B. chinensis* (Supplementary Fig. 23a) and *H. vulgare* (Supplementary Fig. 24a, b), as well as underwater plants such as *M. pteropus* (Fig. 3c, d) and *I. aquatica* (Supplementary Fig. 25a). These findings highlight the water-resistant properties of the core-shell MNP, enabling its versatile application on wet leaf surfaces across both terrestrial and underwater plant species. Moreover, the micropores gradually disappeared after microneedle removal, leaving no visible phenotypic changes in the treated leaf area, consistent with prior microneedle studies in plants (Supplementary Fig. 26)[24,37].

The core-shell MNP with a C10:PCL ratio of 1:3 in the shell was selected for subsequent experiments as it exhibited excellent water resistance while maintaining rapid cargo release upon insertion into plant tissues. This formulation enabled direct application to aquatic plants in shallow water, exhibiting stable adhesion to leaf surfaces for over 1 h under both static and flowing water conditions (Supplementary Movie 2 and 3). For deeper or extended underwater applications, the core-shell MNP can alternatively be enclosed in a protective blister pack (Supplementary Fig. 27a). Inspired by the design of medicinal tablet packaging, the blister pack provides a dry environment to preserve MNP prior to use. This simple and cost-effective design, estimated at a fraction of a dollar per unit (Supplementary Table 2), ensures long-term stability by minimizing contact between the MNP and water before application (Supplementary Fig. 27b). The sealing and opening processes are user-friendly and reversible, allowing the blister pack to be opened underwater, exposing the MNP for direct application on aquatic plants (Supplementary Fig. 27a). The delayed cargo burst release, afforded by the core-shell formulations, rendered the MNP water-resistant in the first several minutes of water exposure, providing sufficient time to apply the MNP on aquatic plants before cargo release. After delivery of the encapsulated cargo, MNP was removed from water (Supplementary Fig. 27c). Alternatively, the MNP residue, composed of biodegradable PCL, may also be left underwater to degrade naturally without toxicity effects to the aquatic ecosystems[38].

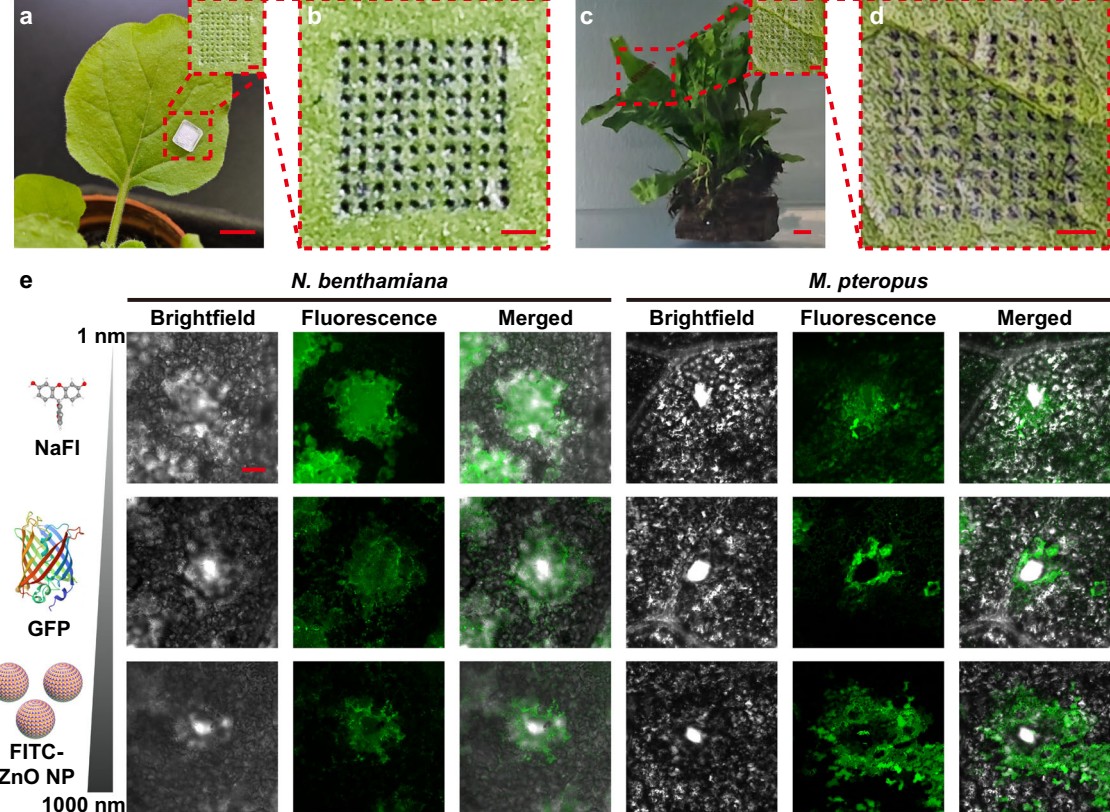

**Fig. 3 | Insertion and cargo delivery by core-shell MNP on live plants. a** Images of core-shell MNP inserting on *N. benthamiana* (scale bar, 1 cm, out of the inset; 1 mm, in the inset) and (**b**) trypan blue staining images of the micropores (scale bar, 1 mm). Imaging was repeated three times independently with similar results. **c** Images of core-shell MNP inserting on *M. pteropus* (scale bar, 1 cm, out of the inset; 1 mm, in the inset) and (**d**) trypan blue staining images of the micropores (scale bar, 1 mm). Imaging was repeated three times independently with similar results.

**e** Representative confocal images showing green fluorescence in the leaves of *N. benthamiana* and *M. pteropus* after treatment with different cargo-loaded core-shell MNPs (scale bar, 100 μm). Cargo: NaFl, GFP and FITC-ZnO NP. Imaging was repeated three times independently with similar results. The 3D image of NaFl was obtained from PubChem (https://pubchem.ncbi.nlm.nih.gov/compound/10608). The 3D image of GFP was obtained from the RCSB PDB (RCSB.org) of PDB ID 1GFL[39].

Trypan blue staining assay was performed to verify the successful insertion of core-shell MNP into plant tissues. Micropores induced by core-shell MNP application were stained blue on both *N. benthamiana* (terrestrial plant) and *M. pteropus* (aquatic plant), demonstrating effective MNP penetration in wet environments (Fig. 3b, d, and Supplementary Fig. 28). The core-shell MNP could be applied on either the adaxial or the abaxial surface of the leaf (Supplementary Fig. 28), showcasing its versatility for cargo delivery over syringe-mediated infiltration, which is mostly effective when applied to the abaxial leaf surface.

To evaluate the multiscale delivery capability of core-shell MNPs, we tested fluorescent cargoes of varying sizes: fluorescein sodium salt (NaFl, ~1 nm), green fluorescent protein[39] (GFP, several nm) and FITC-ZnO nanoparticles (hundreds of nm). FITC-ZnO nanoparticles were synthesized by labeling ZnO nanoparticles with fluorescein isothiocyanate (FITC) using an aminosilane linker to facilitate visualization of cargo in plant tissues (Supplementary Fig. 29a). Comprehensive characterization, including size and zeta potential measurements, elemental analysis, and electron microscopy imaging, confirmed successful FITC attachment to ZnO nanoparticles (Supplementary Fig. 29b–h). Core-shell MNPs loaded with NaFl, GFP and FITC-ZnO nanoparticles were applied to plant leaves and rinsed to remove residual cargoes. Strong green fluorescence was observed at all MNP-induced micropores in both terrestrial and aquatic plants, confirming efficient cargo delivery via the core-shell MNPs (Fig. 3e, and Supplementary Fig. 30a, b). Importantly, fluorescence intensity did not vary significantly between edge and center regions of the patch,

demonstrating uniform microneedle performance across the entire device in terms of both insertion efficiency and delivery consistency (Supplementary Fig. 30c, d). These results demonstrate the versatility of core-shell MNPs in delivering cargoes ranging from 1 nm to several hundred nanometers, enabling effective delivery of multiscale cargoes to both aboveground and aquatic plants.

Preserving cargo bioactivity is critical for the effective delivery of biomolecules and microorganisms. We systematically evaluated three representative cargo types: protein (GFP), nucleic acid (plasmid DNA encoding 35S:GFP), and live microorganisms (*A. tumefaciens*). No significant differences in recovery were observed for GFP (Supplementary Fig. 31) or plasmid DNA (Supplementary Fig. 32) across the fresh cargo solution, core MNP and core-shell MNP formulations. In the case of *A. tumefaciens*, most viability loss occurred during the dehydration step in core MNP preparation (Supplementary Fig. 33a, b), consistent with previous findings[40]. Following core-shell microneedle encapsulation, ~74% of viable bacteria could be recovered (Supplementary Fig. 33c), confirming preservation of microbial bioactivity. Bacterial viability remained high even after 7 days of storage, with over 60-fold improvement compared to previous MNP systems (Supplementary Fig. 33d, e)[37].

## Delivery of plasmid DNA to living plants by core-shell MNP

Building on the demonstrated capability of MNP to deliver fluorescent cargoes into tissues of living plants, we next evaluated their potential for delivering bioactive cargoes relevant to plant engineering (Fig. 4). Core-shell MNPs were loaded with GFP-encoding plasmid DNA to

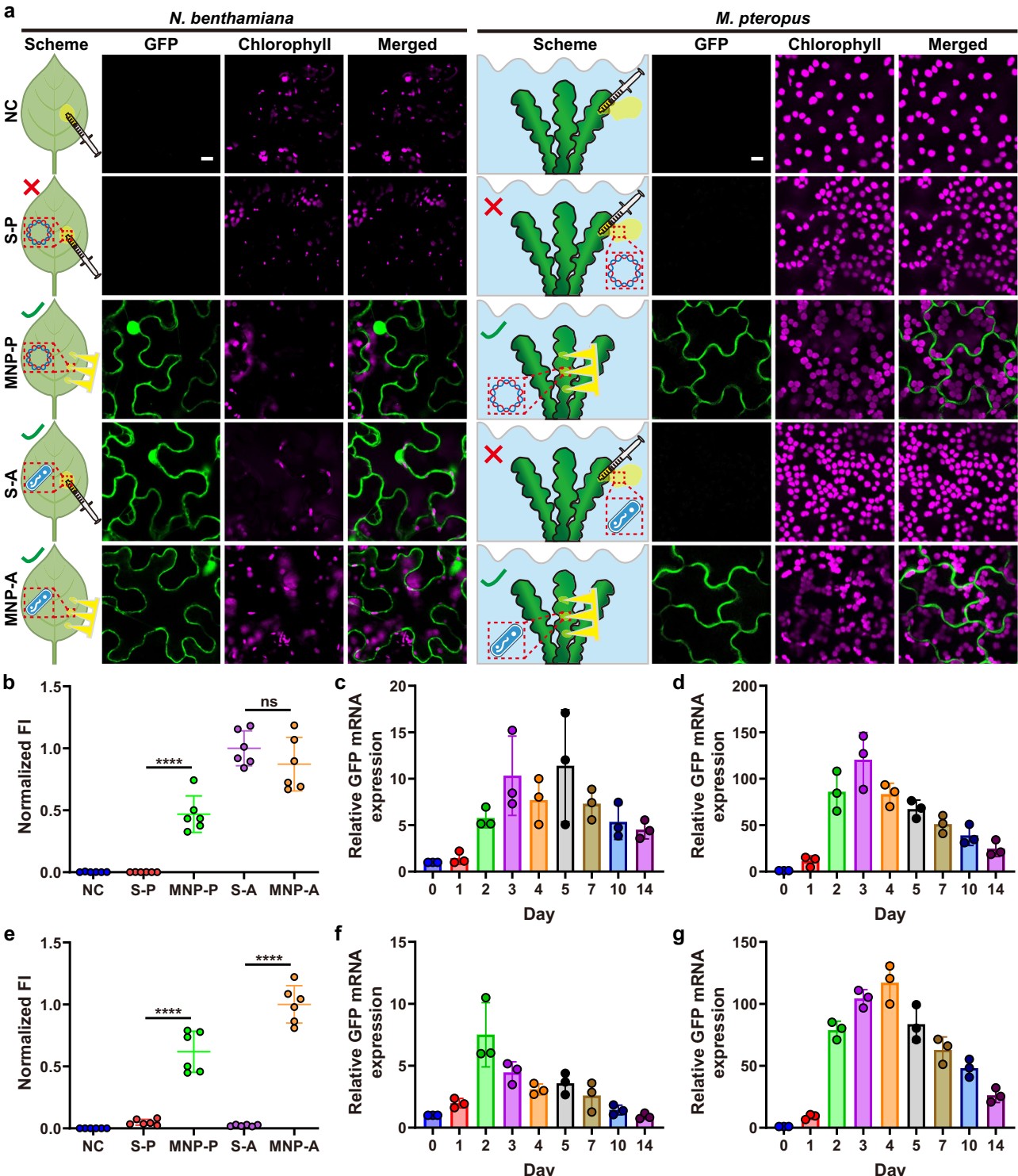

**Fig. 4 | Delivery of GFP-encoding plasmid DNA and active *A. tumefaciens* by core-shell MNP. a** Representative confocal images of *N. benthamiana* and *M. pteropus* (scale bar, 10 μm). Schemes are partially created in BioRender. Lew, T.T.S. (https://BioRender.com/6hf5u8f). **b** Quantification of GFP fluorescence intensity of *N. benthamiana* groups and time profile of GFP mRNA expression, as monitored by RT-qPCR, for (**c**) MNP-P and (**d**) MNP-A treatment. **e** Quantification of GFP fluorescence intensity of *M. pteropus* groups and time profile of GFP mRNA expression, as monitored by RT-qPCR, for (**f**) MNP-P and (**g**) MNP-A treatment. NC: plants treated with infiltration buffer; S-P: plants treated with syringe infiltration of GFP-encoding plasmid DNA; MNP-P: plants treated with plasmid-loaded core-shell MNP. S-A: plants treated by syringe infiltration of *A. tumefaciens*; MNP-A: plants treated by *A. tumefaciens*-loaded core-shell MNP. Data of (**b**–**g**) represent mean ± s.d. (*n* = 6 independent biological replicates, **b**, **e**; *n* = 3 independent biological replicates, each with 3 technical replicates, **c**, **d**, **f**, **g**). Statistical analysis was performed by one-way ANOVA with Tukey's multiple comparisons test. *ns* (not significant), *P* ≥ 0.05, and ****P* < 0.0001. Source data are provided as a Source Data file.

assess their suitability as a gene delivery platform for plants. Syringe-mediated pressurized infiltration of the plasmid DNA (S-P) was used as a control treatment. GFP fluorescence was observed in the mesophyll of terrestrial plants (*N. benthamiana* and *B. chinensis*) and aquatic plants (*M. pteropus* and *I. aquatica*) 3 days after MNP-mediated plasmid delivery (MNP-P), indicating successful transgene delivery and subsequent GFP expression (Fig. 4a). In contrast, no GFP fluorescence was observed in the S-P group or the untreated negative control (NC) group. This is consistent with previous reports that naked DNA cannot internalize into plant cells following syringe-mediated infiltration[41,42]. Quantitative analysis of mean fluorescence intensity confirmed significantly higher GFP expression in the MNP-P group compared to the S-P and NC groups (Fig. 4b, e). Interestingly, *M. pteropus*, unlike *N. benthamiana*, exhibited predominantly cytoplasmic fluorescence with minimal nuclear localization under confocal microscopy. This difference likely arises from the distinct cellular architecture and physiological environment of *M. pteropus* under aquatic growth conditions, which may alter nuclear permeability, cytoplasmic viscosity, and GFP folding or maturation efficiency[43–45].

Western blot analysis corroborated our findings, showing a GFP band exclusively in the MNP-P group, demonstrating successful MNP-mediated transgene delivery and functional protein expression in both terrestrial and aquatic plants (Supplementary Fig. 34). The persistence of GFP in *N. benthamiana* and *M. pteropus* following MNP-mediated plasmid delivery was further quantified at mRNA level using RT-qPCR. In *N. benthamiana*, relative GFP mRNA expression reached its maximum between 3–5 days post-MNP application, with sustained expression up to 14 days, indicating that core-shell MNP-mediated delivery induced transient GFP expression (Fig. 4c). Similarly, in *M. pteropus*, maximal GFP expression was observed 2 days after MNP application before returning to baseline levels by day 14 (Fig. 4f). These results suggested that GFP expression in both terrestrial and aquatic plants was transient, indicating that transgenes delivered by MNP did not integrate into the plant genome, similar to previous observations with nanoparticle-mediated transgene delivery[41,42]. Such a transient transformation approach may be advantageous in plant biotechnology applications where transgene integration is undesirable, such as transgene-free plant genome editing for generating genetically modified organisms (GMOs)[46,47].

## Delivery of viable microorganisms by core-shell MNP

Delivering active microorganisms with MNPs presents technical challenges, as the formulation and fabrication processes can potentially compromise microbial viability[48]. To investigate whether core-shell effervescent MNPs can preserve the viability and activity of microorganisms, live *A. tumefaciens* were encapsulated in the MNP core to produce *A. tumefaciens*-loaded MNP (MNP-A). *A. tumefaciens* was chosen as the model microorganism due to its importance in plant genetic engineering as an interkingdom DNA transfer vector[49]. A GFP-encoding plasmid was introduced into *A. tumefaciens*, and GFP expression in host plants could be monitored to assess its activity. SEM imaging showed that the bacterial morphology was preserved compared to untreated cells, confirming the viability of A. tumefaciens in MNP-A (Supplementary Fig. 35a, b). To visualize bacterial transfer into plant tissues, *A. tumefaciens* was stained with SYTO 9, a green-fluorescent viable dye. Strong green fluorescence was observed around the microneedle-induced micropores, in contrast to blank MNP controls (Supplementary Fig. 35c). Quantification of viable *A. tumefaciens* recovered from plant tissues post-delivery revealed ~$2.7 \times 10^4$ CFU in *N. benthamiana* and ~$2.3 \times 10^4$ CFU in *M. pteropus*, confirming successful delivery of live microorganisms via MNP-A (Supplementary Fig. 35d, e).

Syringe-mediated *A. tumefaciens* infiltration (S-A), a common delivery technique in plant genetic engineering[50], served as a control technique. In *N. benthamiana*, strong GFP expression was observed in both S-A and MNP-A treatments (Fig. 4a), confirming that the core-shell MNP maintained the viability and gene transfer functionality of encapsulated *A. tumefaciens*. However, in aquatic *M. pteropus*, only MNP-A treatment resulted in GFP fluorescence, with no expression detected in the S-A group, likely due to bacterial leakage to the surrounding water during syringe infiltration (Fig. 4a). These results demonstrate that syringe infiltration is unsuitable for underwater cargo delivery, whereas core-shell MNPs offer broader versatility across different environments. Similar findings in *B. chinensis*, *H. vulgare* and *I. aquatica* further highlight the species-agnostic applicability of the core-shell MNP platform and its ability to maintain the bioactivity of encapsulated microorganisms for plant engineering (Supplementary Figs. 23-25).

Quantitative analysis of relative GFP expression in terrestrial plants showed that MNP-A delivery achieved ~90% of the expression level observed with S-A treatment, demonstrating that the core-shell effervescent MNP is an efficient and competitive platform for *A. tumefaciens* delivery (Fig. 4b). Anti-GFP Western blot confirmed the successful GFP expression following *A. tumefaciens* delivery by core-shell MNP (Supplementary Fig. 34). RT-qPCR analysis corroborated these results at the mRNA level, showing 238-fold GFP mRNA increase in the S-A treatment of *N. benthamiana* (Supplementary Fig. 36). In comparison, MNP-A treatment resulted in ~120-fold higher mRNA expression compared to the NC group in both *N. benthamiana* (Fig. 4d) and *M. pteropus* (Fig. 4g).

As a comparison to MNP-P and MNP-A, the non-responsive core-shell MNP (shell: PCL; core: $HA/KHCO_3$) was evaluated for the delivery of plasmid DNA and *A. tumefaciens* (Supplementary Fig. 37a). This formulation exhibits slow-release kinetics and lacks water-triggered effervescence (Fig. 2c). As a result, it showed poor delivery performance with no detectable GFP fluorescence in either *N. benthamiana* or *M. pteropus* (Supplementary Fig. 37b). Western blot analysis further confirmed the absence of GFP expression (Supplementary Fig. 38). Conventional MNPs, such as dissolvable and hydrophobic MNP, were also evaluated for comparison. Dissolvable MNPs, such as the core-only MNP in "No shell" group of Fig. 2g, rapidly disintegrated in water under 10 sec, preventing successful insertion into *M. pteropus* tissues. Attempts to temporarily protect them using blister packs failed to allow timely insertion after exposure to water, highlighting the need of a hydrophobic MNP shell design for underwater use. Hydrophobic MNPs, such as PCL-based, required organic solvents for fabrication, which compromised the integrity of sensitive biomolecular and microbial cargoes, resulting in poor delivery performance (Supplementary Fig. 39). These findings emphasize the necessity of water-responsive core-shell design, which enables efficient and versatile delivery of both biomolecules and live microorganisms, facilitating transgene expression in both terrestrial and submerged aquatic plants (Supplementary Fig. 40).

## Delivery of chemiluminescent reporters and stress-tolerance genes

To further validate the versatility and robustness of the water-responsive MNP platform, we employed firefly luciferase-encoding plasmid DNA as an alternative reporter gene (Supplementary Fig. 41). As chemiluminescence does not require external excitation, it offers a lower background signal than fluorescence-based systems. The luciferase plasmid was first introduced into *A. tumefaciens* and infiltrated into *N. benthamiana* (S-A group, Supplementary Fig. 41a). After 3-day incubation, luciferin was applied to the same region to activate the luciferase reporter system. strong chemiluminescence signal was observed at the infiltration area (Supplementary Fig. 41b), confirming successful gene expression. Chemiluminescence signals were also detected in the MNP-A (*A. tumefaciens*-loaded MNP) and MNP-P (plasmid-loaded MNP) groups in both *N. benthamiana* (Supplementary Fig. 41b) and *M. pteropus* (Supplementary Fig. 41c, d). In contrast, the

S-A group in *M. pteropus* failed to produce detectable signal, consistent with earlier findings with GFP-based reporters. These results further demonstrate broad applicability of the MNP platform for efficient gene and microorganism delivery in both terrestrial and aquatic plants.

Beyond delivering reporter genes, the MNP platform was further applied to deliver functional genes with physiological relevance to submerged aquatic plants. Specifically, core-shell MNPs were loaded with AoCLCf, a salt-responsive gene recently identified to enhance salt tolerance in *A. thaliana*[51]. As a member of the chloride channel family from the mangrove species *Avicennia officinalis*, AoCLCf plays a critical role in Cl⁻ transport and ion homeostasis under salt stress. However, its function or relevance in aquatic plants remains unexplored due to lack of compatible delivery systems underwater. We used the water-responsive, core-shell MNPs to deliver GFP-tagged AoCLCf into aquatic *M. pteropus* under submerged conditions (Fig. 5). RT-qPCR analysis (Fig. 5b) and confocal microscopy images (Supplementary Fig. 42) confirmed successful gene expression in the MNP-P and MNP-A groups, whereas syringe-infiltrated controls (S-P and S-A) showed negligible GFP expression.

As a freshwater species, *M. pteropus* is highly susceptible to salt stress. Upon exposure to 120 mM NaCl, control plants (PC) exhibited characteristic signs of stress, including darkened, unhealthy leaves (Fig. 5a) In contrast, plants treated with MNP-delivered AoCLCf (MNP-P and MNP-A) displayed visibly improved phenotypes, while syringe-infiltrated groups showed little to no rescue. Quantitative measurements of Chlorophyll a and b (Chla and Chlb) content and photosynthetic performance confirmed that MNP-mediated AoCLCf delivery mitigated salt stress in *M. pteropus* (Fig. 5c–f). Furthermore, reactive oxygen species (ROS) accumulation, a marker of oxidative stress, was also significantly reduced in the MNP-treated groups (Fig. 5g, h). Together, these results demonstrate the unique potential of water-responsive MNPs for delivering functional genes into submerged aquatic plants, enabling physiological trait modification in systems previously inaccessible using conventional delivery techniques. This expands the application of the MNP platform beyond reporter gene studies, highlighting its potential for engineering agriculturally and ecologically relevant traits in underwater plant species.

### Delivery of plant hormones by core-shell MNPs

To further assess the versatility of the core-shell MNP delivery platform, we investigated its potential for hormone delivery to enhance plant growth. While conventional spraying methods are suitable for aboveground plants, hormone delivery to submerged aquatic plants faces significant challenges due to rapid dispersion and dilution into the surrounding water, resulting in inefficient and untargeted delivery. To demonstrate the feasibility of MNP-assisted hormone delivery, indoleacetic acid (IAA), a plant growth-promoting hormone, was encapsulated within core-shell MNPs. The microneedles were then applied to cut stems of *I. aquatica* growing underwater. Root induction was observed at stem base of the aquatic plant, indicating IAA-triggered cell differentiation following MNP-assisted hormone delivery (Supplementary Fig. 43a). Compared to the untreated plants, MNP-mediated IAA delivery resulted in accelerated root growth (Supplementary Fig. 43b). Additionally, MNP-treated stems exhibited a significantly higher number of roots compared to untreated stems, with an average of multiple roots emerging within 3-4 days post-treatment (Supplementary Fig. 43c). These findings demonstrate the successful delivery of IAA to aquatic plants using the core-shell MNP platform, effectively promoting cell differentiation and root emergence which are key indicators of plant growth and development.

### Core-shell MNP is biocompatible with living plants

The biocompatibility of core-shell MNP was evaluated using propidium iodide (PI) staining to visualize dead cells and by quantifying plant photosynthetic performance following MNP application. PI staining revealed a limited number of cells with compromised membranes around the microneedle-treated area, indicating minimal impact on plant cell viability (Supplementary Fig. 44). As a positive control, SDS-treated leaves exhibited gradual cuticle destruction over a 7-day observation period, while no significant damage was found in buffer-treated and MNP-treated areas (Fig. 6a). Consistent with this, quantitative imaging of the maximum photosynthetic quantum efficiency of Photosystem II (Fv/Fm) demonstrated compromised photosynthetic ability exclusively in the SDS-treated area. Similar findings were observed in *M. pteropus*, where SDS treatment resulted in visible leaf darkening due to cuticle damage and chlorophyll loss. In contrast, MNP treatment had negligible effects on both phenotypic appearance and Fv/Fm profile.

Further quantification of photosynthetic parameters, including Fv/Fm, non-photochemical quenching (NPQ), Photosystem II operating efficiency (qP) and electron transport rate (ETR), revealed no significant differences between MNP-treated and buffer-treated plants over 7 days post-treatment (Fig. 6b). Chlorophyll content, a key indicator of plant health, also remained stable in the MNP-treated group, in contrast to the continuous decline observed in SDS-treated *N. benthamiana* and *M. pteropus* (Fig. 6c, d). Additionally, transient OJIP curves exhibited characteristic O, J, I and P steps in both MNP-treated and buffer-treated plants, confirming the normal functioning of the photosynthetic system after MNP treatment (Fig. 6e, f). Collectively, these observations suggest that core-shell MNP possesses excellent biocompatibility with minimal impact on plant photosynthetic efficiency and leaf cell viability.

## Discussion

In this work, we have developed a water-responsive MNP for programmable delivery of biomolecules and active microorganisms to both terrestrial and aquatic plants. The core-shell MNP feature a spatially compartmentalized structure containing effervescence reaction precursors. Specifically, $KHCO_3$ was confined to the core, while a poorly soluble fatty acid was incorporated into the shell, ensuring that the effervescent reaction is activated only upon hydration in plant tissues to trigger cargo release. Inspired by the effervescent reactions in medicinal tablets, we harnessed reaction-diffusion dynamics to rationally engineer core-shell microneedles with programmable cargo release kinetics in wet environments, ranging from minutes to days. Our results indicated that multiscale biomolecules, ranging from ~1 nm to 800 nm, could be efficiently loaded in the microneedle core and successfully delivered into plant tissues. We showed that microneedles can be designed to deliver plasmid DNA into living plant tissues, enabling transient protein expression as confirmed at the mRNA level by RT-qPCR and at the protein level through confocal microscopy and Western blot analysis. The resulting protein expression in mature leaves is strong enough for detection by Western blot, overcoming limitations of low delivery efficiency often encountered in nanomaterial-based gene delivery[52–54]. In such platforms, strong electrostatic interactions may limit the availability of nucleic acids to the plant's transcription machinery, resulting in weak protein expression[53]. In contrast, our core-shell microneedle platform encapsulates cargo through physical entrapment, ensuring its stability while enabling controlled release. Once inside plant cells, an effervescent reaction rapidly accelerates cargo release from microneedles, enhancing the accessibility of nucleic acids for efficient protein expression.

We demonstrated that the water-responsive core-shell MNP platform enables efficient, controlled and biocompatible delivery of diverse cargoes to both terrestrial and aquatic plants. This system supports the delivery of a broad spectrum of payloads, from nanometer-scale biomolecules to micrometer-scale living microorganisms, while maintaining their bioactivity. Its applicability to both leaves and stems, across species with varying leaf textures, cuticle thicknesses, and growth environments, including both terrestrial and

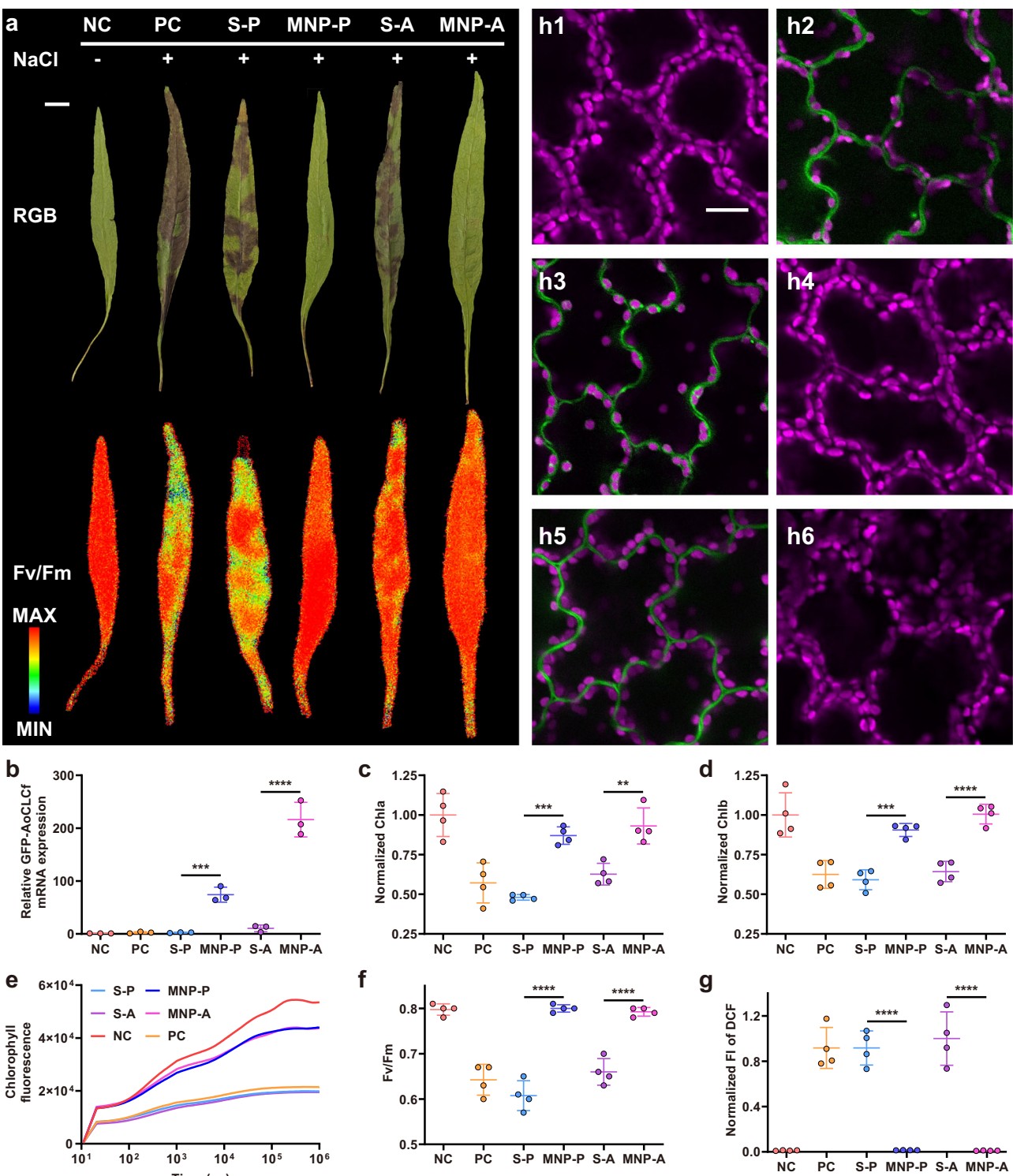

**Fig. 5 | Delivery of GFP-AoCLCf-encoding plasmid DNA and active *A. tumefaciens* by core-shell MNP. a** Quantitative imaging of maximum photosynthetic quantum efficiency (Fv/Fm) of plant leaves after treatments (scale bar, 1 cm). **b** mRNA expression of GFP-AoCLCf as monitored by RT-qPCR. Quantification of (**c**) Chla and (**d**) Chlb of leaves after treatments. **e** OJIP transient curves of leaves after different treatments. **f** Fv/Fm quantification of leaves after treatments. **g** Quantification FI of DCF and the representation confocal images of DCFH-DA-stained leaves for (h1) NC, (h2) PC, (h3) S-P, (h4) MNP-P, (h5) S-A and (h6) MNP-A (scale bar, 20 μm). NC: plants kept in water and treated with infiltration buffer; PC: plants exposed in NaCl solution and treated with infiltration buffer; S-P: plants

exposed in NaCl solution and treated with syringe infiltration of GFP-AoCLCf-encoding plasmid DNA; MNP-P: plants exposed in NaCl solution and treated with plasmid-loaded core-shell MNP. S-A: plants exposed in NaCl solution and treated by syringe infiltration of *A. tumefaciens*; MNP-A: plants exposed in NaCl solution and treated by *A. tumefaciens*-loaded core-shell MNP. Data of (**b**), (**c**), (**d**), (**f**) and (**g**) represent mean ± s.d. (*n* = 3 independent biological replicates, each with 3 technical replicates, **b**; *n* = 4 independent biological replicates, **c**, **d**, **f** and **g**). Statistical analysis was performed by one-way ANOVA with Tukey's multiple comparisons test. **P < 0.01, ***P < 0.001, and ****P < 0.0001. Source data are provided as a Source Data file.

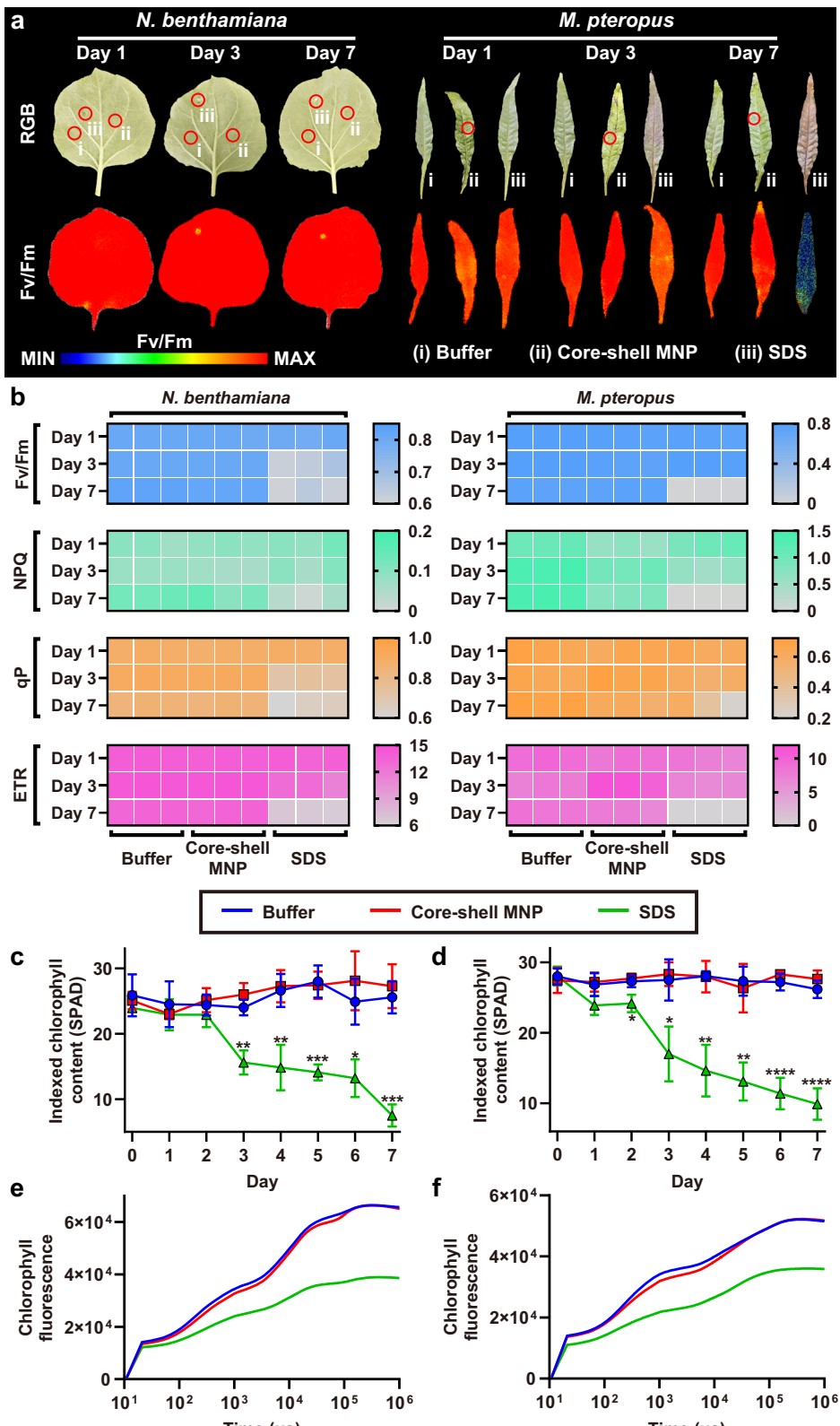

**Fig. 6 | Biocompatibility of core-shell MNP in living plants. a** Quantitative imaging of Fv/Fm of plant leaves over 7 days post-treatments. **b** Photosynthetic indices (Fv/Fm, NPQ, qP and ETR) of plant leaves under the 7-day treatments. Each treatment group consisted of 3 independent replicates. **c, d** Quantification of chlorophyll content (SPAD value) of (c) *N. benthamiana* and (d) *M. pteropus* over 7 days post-treatments (mean ± s.d., *n* = 3 independent biological replicates). **e, f** OJIP transient curves of (**e**) *N. benthamiana* and (**f**) *M. pteropus* after different treatments. Statistical analysis was performed by one-way ANOVA with Tukey's multiple comparisons test. *$P < 0.05$, **$P < 0.01$, ***$P < 0.001$, and ****$P < 0.0001$ compared with the control group. Source data are provided as a Source Data file.

aquatic plants, highlights its broad versatility. Importantly, this platform enables targeted gene and microbial delivery under challenging natural conditions, such as high humidity or full submergence, where conventional techniques such as syringe infiltration, foliar spraying and biolistic bombardment often fail. Furthermore, beyond reporter gene delivery, we demonstrated the delivery of a functional salt-tolerance gene that enhanced stress resilience in *M. pteropus* under saline conditions, a realistic biological challenge for freshwater aquatic plants. These findings illustrate the platform's potential to introduce functional traits in crops and non-model species of ecological significance. Overall, this work establishes a versatile, amphibious microneedle system for in situ, targeted cargo delivery across diverse plant systems. By enabling precise molecular delivery to intact tissues under challenging conditions, the core-shell MNPs complement and expand the current toolbox of plant delivery technologies, opening avenues for basic research and genetic manipulation in agriculture, environmental sustainability, and plant synthetic biology.

These microneedles are not intended to replace existing plant transformation or delivery techniques, but rather to complement and expand their applicability. We demonstrated the compatibility of the microneedles with *Agrobacterium*-mediated transformation by using them to successfully deliver *A. tumefaciens* into aquatic plant tissues in an environment where syringe-mediated infiltration typically fails (Fig. 4a). As such, this work establishes a generalizable and programmable microneedle strategy that enhances the accessibility of genetic and microbial delivery in challenging plant systems, including submerged aquatic species often overlooked in plant biotechnology.

Recent advances in 3D printing of microneedles offer a promising route for improving accessibility to plant researchers and non-experts, while also ensuring reproducible fabrication and scalability[55]. In parallel, commercial-scale microneedle manufacturing is now technically and economically feasible. For example, Beijing CAS Microneedle Technology Ltd. (China) reports a manufacturing capacity of ~6 billion patches per year (http://www.casmn.com/col.jsp?id=138), based on dimensions comparable to this MNP platform, highlighting the translational potential of this technology for broader deployment. Furthermore, the patch-based format of MNPs is inherently compatible with gripper-based actuators, supporting potential integration with robotic platforms for automated application[56]. Coupled with recent developments in agricultural and underwater robotics, this compatibility paves the way for field-scale deployment in both terrestrial and aquatic environments[57].

Additionally, quantification of photosynthetic performance and cell viability underscored the biocompatibility of these microneedles, exerting minimal impact on photosynthetic ability and cellular health of treated plants. Taken together, this work introduces a versatile, amphibious tool for in situ, targeted, and controlled cargo delivery to intact plants, overcoming longstanding barriers in underwater plant biotechnology and establishing a promising platform to advance plant engineering for food security, environmental sustainability, and aquatic ecosystem management.

## Methods
### Materials
Capric acid (C10), lauric acid (C12), myristic acid (C14) and dichloromethane (DCM) were purchased from Tokyo Chemical Industry Ltd. (Japan). Sodium hyaluronate (HA, 90-100 kDa) was purchased from Shanghai Yuanye Bio-Technology Ltd. (China). Poly-ε-caprolactone (PCL, 30-45 kDa) was purchased from Shanghai Aladdin Chemical Reagent Ltd. (China). Green fluorescent protein (GFP) was purchased from Proteintech (Germany). Zinc oxide nanoparticles (ZnO NP) were purchased from Strem Chemicals (USA). Unless otherwise specified, all other reagents were purchased from Sigma-Aldrich (Singapore). MNP mold (ST-10×10-

H300B100P500) was purchased from Micropoint Technologies (Singapore).

### Characterization
Zeta potential and dynamic light scattering (DLS) measurements were performed by Zetasizer Nano ZSP (Malvern). Transmission electron microscopy (TEM) imaging was performed by JEM-2100F field emission electron microscope. SEM imaging and energy-dispersive X-ray spectroscopy (EDS) elemental mapping were performed by JSM-7610FPlus Schottky field emission scanning electron microscope. Fluorescence imaging was performed by Olympus BX51 fluorescence microscope. pH measurements were performed by a pH meter (LAQUA PH2000, HORIBA). X-ray diffraction (XRD) experiment was performed by D8 ADVANCE (Bruker). Fourier transform infrared (FTIR) spectroscopy was performed by VERTEX 70 (Bruker).

Gas chromatography/mass spectrometry (GC/MS) experiments were performed to determine residual DCM or HFIP in core-shell MNP samples. 8 samples were analyzed in total, including 2 extracts from MNP samples and 6 control samples (DCM, MeOH, HFIP, and their mixed samples). Each sample was analyzed once (independent sample, n = 1). The core-shell MNP samples were immersed in MeOH overnight and the supernatants were collected as the extract samples, respectively. All the samples were filtered through 0.22 μm membranes before testing. GC/MS experiments were performed by Agilent 7890B-5977B MSD (column: 19091S-433UI). The system temperature was initially maintained at 50 °C for 3 min, and then increased to 280 °C at the rate of 20 °C/min, and finally kept constant at 280 °C for an additional 2 min. The chromatographic profiles of the MNP extract samples were compared with the controls to determine the presence of residual DCM or HFIP.

### Plant growth and treatments
*B. chinensis*, *H. vulgare* and *I. aquatica* seeds were purchased from NTUC FairPrice Co-operative Ltd. (Singapore). *M. pteropus* plants were purchased from Just Betta (Singapore). Terrestrial and aquatic plants were grown in a controlled growth chamber (FitoClima 600 PLH, Aralab, Portugal) at 23 °C with a 14 h/10 h day/night cycle. *N. benthamiana*, *B. chinensis* and *H. vulgare* were grown in soil (Jiffy Group). 6-week-old *N. benthamiana*, 3-week-old *B. chinensis* and 3-week-old *H. vulgare* were used for experiments. 4-week-old *I. aquatica* and 12-month-old *M. pteropus* were grown in a hydroponic culture with Hoagland nutrient solution. A 1-mL needleless syringe was used to infiltrate the GFP plasmid solution or *A. tumefaciens* suspension from the abaxial side of a leaf. MNP could be inserted from either the adaxial or abaxial side of a leaf, as stated in the main text.

In the hormone delivery experiments, plant stems of ~5 cm long were collected from 4-week-old *I. aquatica*. The leaves were cut and the residual stem was mildly dried at room temperature for 15 min. The bottom of the stem was subsequently immersed into water. Meanwhile, a core-shell MNP encapsulating IAA was inserted into the stem and then removed after 1 hour. Phenotypic changes were quantified over a period of 6 days after treatment.

### Bacteria culture and plasmid extraction
The coding sequence (CDS) of GFP was cloned into a pGreen binary vector, resulting in the construct 35S:GFP (without a nuclear localization signal tag)[58,59]. This construct was then introduced into *A. tumefaciens* strain GV3101:pMP90. *A. tumefaciens* were grown for 1 day under 30 °C in Luria-Bertani broth (LB broth containing 0.1 mg/mL gentamicin, 0.1 mg/mL kanamycin and 0.05 mg/mL rifampicin) and kept at 4 °C before use. Similarly, constructs expressing firefly luciferase[60] (35S:LUC) and GFP-AoCLCf[51] (35S:GFP-AoCLCf) were generated by cloning their respective CDS into the pGreen binary vector and introduced into *A. tumefaciens* according to the same procedure. Plasmid was extracted with the plasmid extraction mini kit

(FavorPrep™) according to the protocol provided by the manufacturer. Plasmid concentration was verified by Nanodrop 2000 (Thermo Fisher Scientific).

## Microneedle preparation

The preparation process of the core MNP and core-shell MNP followed the mold-aiding strategy. Briefly, HA and $KHCO_3$ were dissolved in water to make Solution A. PCL and fatty acid were dissolved in DCM to make Solution B. 100 mg of Solution A was added into the microneedle mold. The solution-mold system was kept in a vacuum chamber (20 mbar) for 15 min and a desiccator for 12 h, successively. After this treatment, the core MNP could be detached from the mold. The core MNP was subsequently immersed in Solution B for 5 sec and then moved out. After 12-h evaporation of DCM, the core-shell MNP would be obtained. DCM could be replaced by hexafluoroisopropanol (HFIP) in the preparation process. A specified amount of cargo could be pre-added in Solution A to prepare cargo-loaded core-shell MNP. Unless otherwise specified, the core MNP was prepared with a formulation of 4:1 (wt%) for HA:$KHCO_3$, and the core-shell MNP was formulated based on a core formulation of 4:1 (wt%) for HA:$KHCO_3$ and a shell formulation of 1:3 (wt%) for C10:PCL.

To prepare the *A. tumefaciens*-loaded core-shell MNP, *A. tumefaciens* culture was centrifuged at $7000 \times g$ for 3 min. The pellet was re-dispersed in fresh LB broth (to prolong bacterial viability in MNP)[61] and then mixed with HA and $KHCO_3$ as Solution A (HA 80 mg/g; $KHCO_3$ 20 mg/g; *A. tumefaciens* - $10^9$ CFU/g). To prepare the plasmid-loading core-shell MNP, 90 mg of prepared plasmid solution (~50 ng/µL), 8 mg of HA and 2 mg of $KHCO_3$ were mixed together as Solution A. To prepare the IAA-loading core-shell MNP, 1 mg of IAA, 8 mg of HA and 2 mg of $KHCO_3$ were dissolved in 89 mg of water as Solution A. The following steps were the same as previously described to prepare cargo-loaded MNPs. All the MNPs were stored in a dry box (25 °C, 30% RH) before use. Preparation of core-shell MNP with cargo in microneedle tips and hydrophobic MNP were described in Supplementary Method 1 and 2.

## Preparation of underwater MNP platform

Firstly, the edges of the core MNP were trimmed. The core-shell MNP was subsequently prepared from the trimmed core MNP (including microneedles and the flat base) following the same procedure above. Finally, the core-shell MNP was stuck onto the cellulose adhesive layer. Now it is ready for usage in the shallow water applications. This system could be further sealed in a blister pack for deeper or extended underwater tasks.

## Synthesis of potassium decanoate (K-C10)

A mixture containing 1 mmol of C10 and 10 mL 0.1 mmol/mL of KOH aqueous solution was prepared. The mixture was stirred under room temperature for 30 min to form a clear solution. The solution was lyophilized to obtain K-C10.

## Synthesis of FITC-ZnO NP

FITC-ZnO NP was synthesized according to a previous report with slight modifications[62]. Briefly, 1 g ZnO NP was dispersed in 100 mL of 1% KOH ethanol solution. 2 mL of (3-aminopropyl)triethoxysilane was added to the dispersion above followed by stirring at 500 rpm, 70 °C for 2 h. The dispersion was centrifuged ($7000 \times g$, 15 min) and the pellet was collected. ZnO-$NH_2$ NP was obtained after washing by ethanol for 3 times and lyophilized. 20 mg of lyophilized ZnO-$NH_2$ NP was dispersed in 10 mL 1% $NaHCO_3$ solution. Then, the dispersion was mixed with 1 mL of 2 mg/mL fluorescein isothiocyanate (FITC) DMSO solution and stirred at 25 °C under 500 rpm. After 2 h, the mixture was centrifuged ($7000 \times g$, 15 min) and the pellet was collected. FITC-ZnO NP was obtained after washing with DI water for 3 times and lyophilized.

## Extraction of plant apoplastic fluid

Plant apoplastic fluid was extracted from *N. benthamiana* as described previously[63]. Briefly, 10 *N. benthamiana* leaves were immersed in 20 mM MES (containing 2 mM $CaCl_2$ and 100 mM NaCl, pH 6.0) and kept under vacuum for 30 min. The infiltrated leaves were then centrifuged at 4 °C under $2000 \times g$ for 10 min. The apoplastic fluid was collected from the bottom of centrifuge tubes after the centrifugation process.

## Release of fluorescent cargo from MNP

Rhodamine B (RhB), fluorescein sodium salt (NaFl) and Nile red were used as the model cargoes to study in vitro cargo release kinetic[64]. As a typical procedure, a piece of cargo-loaded core-shell MNP was immersed in 4 mL of medium at 25 °C. 0.2 mL of supernatant was collected at the designed time points, while 0.2 mL of fresh medium was then added to the stock medium to maintain the same total volume. For RhB and NaFl, the fluorescence intensity (FI) of collected supernatant could be directly obtained with a microplate reader (BioTek Synergy H1). The excitation (Ex.) and emission (Em.) wavelengths were 510 and 579 nm for RhB, and 480 and 520 nm for NaFl, respectively. For Nile red, the collected supernatant (0.2 mL) was first diluted with ethanol (1.8 mL) and then incubated for 1 hour at 25 °C before FI measurement (Ex. 510 nm, Em. 640 nm) with the same microplate reader.

RhB was also used as a model cargo to study MNP-mediated delivery into leaves. Each RhB-loaded core-shell MNP sample (loading in whole MNP or only in microneedle tips) was inserted on a piece of *N. benthamiana* leaf tissue (1 cm × 1 cm), which was kept in a chamber (25 °C, 90%RH) throughout the experiment. The high relative humidity was employed to prevent dehydration of the leaf tissue. At the designed time points, the MNP was separated from leaf tissue. The MNP was then immersed in water to extract residual RhB, and the treated leaf tissue was homogenized in 0.5 mL of water. The supernatant samples from extracted MNP and leaf homogenate were collected for fluorescence measurement. Fluorescence intensities (Ex. 510 nm, Em. 579 nm) were obtained with the same microplate reader described above.

Phase I (slow release) and Phase II (burst release) were defined to study the delay-release property of MNP. The release behavior of core MNP was dissolution-based, which was described as burst release (Phase II). The release behavior of core-shell MNP with the pure PCL shell was diffusion-based, which was described as slow release (Phase I). For systematic analysis of other MNPs, cargo release phases were defined as Phase I for release rates ≤3%, with the remaining classified as Phase II.

## Characterization of mechanical strength of MNP

Mechanical strength of MNP was analyzed with Instron 5848 Micro-tester according to a previous report[32]. Briefly, the ultraviolet (UV) curing adhesive (Shenzhen Tegu New Material Ltd, China) was added to fill the base part of the MNP, and was subsequently adhered onto a place of glass. A 10-second UV irradiation (365 nm, 100 W) was performed to crosslink the UV curing adhesive. This treatment was done to avoid the unexpected deformation of MNP base in the mechanical strength test. The prepared MNP sample was then kept between two parallel stainless-steel plates of the microtester. The axially compression speed of the upper steel plate was set as 0.5 mm/min. The point of 0 µm was recorded when the detected force achieved 0.01 N. The force-displacement curve was plotted in the range of 0 - 200 µm.

## Consistency evaluation across all microneedles in core-shell MNP

The NaFl-loaded core-shell MNP was used for the evaluation. The MNP was inserted on the leaf of *N. benthamiana* or *M. pteropus*. After 10 min, the MNP was removed from the leaf. The micropores were left on the

leaf and shone by 395 nm light (35 W, InvaCarDan flashlight). Green fluorescence could be observed at each micropore. Quantification of fluorescence intensity was then performed on each micropore by ImageJ.

### GFP recovery post-microneedle encapsulation

Firstly, 2 µg of GFP was loaded in each core MNP. The core MNP was then coated with C10/PCL shell (C10:PCL = 1:3) to form the core-shell MNP. To evaluate GFP recovery, each MNP sample was extracted in 1 mL of water. 2 µg/mL of fresh GFP solution and boiled GFP solution (deactivated GFP) were included as control groups. Fluorescence intensities (FI, Ex. 470 nm, Em. 520 nm) of all the samples were obtained with a microplate reader (BioTek Synergy H1). GFP recovery was quantified based on the calibration curve.

### GFP-encoding plasmid DNA recovery post-microneedle encapsulation

4500 ng of plasmid was first loaded in each core MNP. The core MNP was then coated with C10/PCL shell (C10:PCL = 1:3) to form the core-shell MNP. Each MNP sample was extracted in 100 µL of water. 45 ng/µL of fresh plasmid and deactivated plasmid (treated by deoxyribonuclease I) were included as control groups. Agarose gel electrophoresis was performed to analyze the plasmid samples. 2% (w/v) agarose gel, stained by FloroSafe DNA Stain (1st BASE), was used for the study. Electrophoresis was conducted at 100 V for 30 min. DNA bands were imaged by Amersham ImageQuant 800 under UV illumination.

### Evaluation of bacterial viability

A total of 100 µL of the *A. tumefaciens* solution was diluted $10^6$-fold in saline. Then 100 µL of this liquid was added and spread on the LB agar plate (containing 0.1 mg/mL gentamicin, 0.1 mg/mL kanamycin and 0.05 mg/mL rifampicin). The bacterial colonies were counted after 3-day incubation under 30 °C.

To evaluate bacterial viability of *A. tumefaciens*-loaded MNP, each core MNP or core-shell MNP sample (initially containing the same amount of *A. tumefaciens* as 100 µL of *A. tumefaciens* solution described above). The freshly prepared core MNP and core-shell MNP were used in the test. Besides, the core-shell MNP stored for 7 days (25 °C, 30%RH) was also involved. To quantify the viable *A. tumefaciens*, each MNP sample was extracted in 10 mL of saline and then diluted 10-fold. A total of 100 µL of the liquid was spread on the same LB agar plate above, followed by the same incubation and colony counting procedure.

### Delivery of viable *A. tumefaciens* from MNP-A to plant tissues

To prepare SYTO 9-stained MNP-A, 1 µL of 5 mM SYTO 9 was added to the *A. tumefaciens* solution before mixing with HA and $KHCO_3$. The rest of procedures were the same as that in preparing *A. tumefaciens*-loaded core-shell MNP. The prepared SYTO 9-stained MNP-A was then inserted on the leaf of *N. benthamiana* or *M. Pteropus*. After 10 min, the MNP-A was removed and the leaf was observed by the CLSM system (FLUOVIEW FV3000, Olympus).

To quantify the amount of viable *A. tumefaciens* delivered from MNP-A into the plant tissue, MNP-A was inserted on the leaf of *N. benthamiana* or *M. pteropus*. After 30 min, the MNP-A was removed and the leaf surface was wiped with 75% ethanol. The aim of wiping was to kill *A. tumefaciens* on the leaf surface and retain those that had been delivered into the tissue. After ethanol evaporated, the leaf was homogenized in 0.5 mL of water and diluted properly. Then 100 µL of the liquid was added and spread on the LB agar plate (containing 0.1 mg/mL gentamicin, 0.1 mg/mL kanamycin and 0.05 mg/mL rifampicin). The bacterial colonies were counted after 3-day incubation under 30 °C.

### Delivery of firefly luciferase-encoding plasmid DNA and *A. tumefaciens* in live plants

*N. benthamiana* and *M. pteropus* were involved in the study. A 1-mL needleless syringe was used to infiltrate buffer (NC), the plasmid solution (S-P) or *A. tumefaciens* suspension (S-A) from the abaxial side of a leaf. Water-responsive core-shell MNP loaded with plasmid (MNP-P) or *A. tumefaciens* (MNP-A) were inserted into the leaf. After 3-day incubation, 1 mM D-luciferin sodium salt ("luciferin" for short) solution was infiltrated to the same location of treatments (*N. benthamiana*) or was dropped on the entire leaf (*M. pteropus*). Leaf samples were kept in darkness for 5 min and chemiluminescence signals in leaves were imaged by PlantView100 (Guangzhou BLT-Imaging Ltd).

### Delivery of GFP-AoCLCf-encoding plasmid DNA and *A. tumefaciens* in live plants

*M. pteropus* was used in the study. A 1-mL needleless syringe was used to infiltrate buffer (NC, PC), the plasmid solution (S-P) or *A. tumefaciens* suspension (S-A) from the abaxial side of a leaf. Water-responsive core-shell MNP loaded with plasmid (MNP-P) or *A. tumefaciens* (MNP-A) were inserted into the leaf. The leaf of NC group was kept in fresh water (without adding NaCl). For other groups, the leaves were first kept in fresh water for 2 days, and then transferred into saline (120 mM NaCl) for another 8-day incubation. Then the photosynthetic performance index (Fv/Fm) were analyzed using Closed FluorCam 1300 (Photon System Instruments, Czech Republic). To quantify chlorophyll amount (Chla and Chlb), leaf samples (diameter: 8 mm) were prepared by a hole puncher and extracted by acetone[65]. The liquid samples were measured by a microplate reader (BioTek Synergy H1). To study the ROS level, the leaf samples were cut into small pieces (5 mm × 5 mm) and immersed in 10 µM DCFH-DA (2′,7′-dichlorodihydrofluorescein diacetate) in darkness overnight. Then the leaf samples were rinsed by DI water and observed for DCF (2′,7′-dichlorofluorescein, Ex. 488 nm, Em. 520 nm) and chloroplast (Ex. 640 nm, Em. 670 nm) by the CLSM system (FLUOVIEW FV3000, Olympus). To observe GFP-AoCLCf in leaves, the treated leaf samples were kept in fresh water for 2 days, and then transferred into saline (120 mM NaCl) for one more day of incubation. Then the leaf samples were rinsed by DI water and observed for GFP-AoCLCf (Ex. 488 nm, Em. 520 nm) and chloroplast (Ex. 640 nm, Em. 670 nm) by the CLSM system (FLUOVIEW FV3000, Olympus).

### PI staining assay

After treatment with the core-shell MNP, each plant leaf was immersed in PI solution (0.5 mg/mL) for 10 minutes. The leaves were then rinsed by DI water to remove excess PI on the surface. Fluorescence images of PI-stained leaves were obtained using confocal microscopy. Untreated leaves were used as a negative control.

### Trypan blue staining assay

Core or core-shell MNP was inserted on the plant leaf and removed after 2 min. A micropore would be formed if the microneedle successfully inserted into the leaf. The leaf was then rinsed by water for 30 sec and dried by tissue. 50 µL of 0.4% trypan blue solution was gently dripped on the MNP-treated area of the leaf. After 2 min, the excess trypan blue solution was washed with water. The micropores would be visibly stained blue.

### Confocal laser scanning microscopy (CLSM) imaging and quantitative analysis

CLSM imaging was performed by FLUOVIEW FV3000 (Olympus). Each leaf sample was cut into a small disk and observed on the third day after treatment for GFP (Ex. 488 nm, Em. 520 nm) and chloroplast (Ex. 640 nm, Em. 670 nm). Quantification of GFP fluorescence intensity was performed by ImageJ. Each treatment group included 6 biological replicates.

## Western blot analysis

To confirm the expression of GFP, the treated *N. benthamiana* and *M. pteropus* leaves were lysed with a lysis buffer at 4 °C. Lysis buffer consists of 20 mM Tris-HCl (pH 8.0), 20 mM $K_2HPO_4$, 300 mM NaCl, 10% glycerol supplemented with Mini, EDTA-free protease inhibitor cocktail (Roche). 30 μg of protein lysate was loaded for protein expression analysis. Protein expression and purity were confirmed by SDS-PAGE and western-blot analyses using anti-GFP (Cat # SC-9996, 1:5000) and anti-Mouse (Cat # SC-2005, 1:30000) antibodies (Santa Cruz).

## RNA isolation and quantitative RT-qPCR analysis

RNA was isolated from the treated samples of *N. benthamiana* and *M. pteropus* using Qiagen kit following the manufacturer's instructions. From this, 1 μg of RNA was used, and cDNA was synthesized using Maxima first strand cDNA synthesis kit for RT-qPCR (Thermo Fisher) following the manufacturer's instructions. The RT-qPCR analyses were performed using Biorad Real-Time PCR machine with the following program: 20 sec at 95 °C followed by 40 cycles of 3 sec at 95 °C and 30 sec at 60 °C using SYBR Fast PCR kit (Thermo Fisher). The RT-qPCR data were analyzed using BioRad's CFX Maestro software. The primers used for the RT-qPCR analysis are listed in Supplementary Table 3.

A no-reverse transcriptase (no-RT) control was also included to confirm that the qPCR signals originated from cDNA rather than residual plasmid DNA contamination. Gene expression levels from three biological replicates, each with three technical replicates (total n = 9), were calculated based on ΔΔCT values and represented as relative expression levels (fold change) to constitutively expressed internal control, NbUbiquitin and MpUbiquitin.

## Quantification of plant photosynthetic performance

Approximately 20 μL of water or 10% sodium dodecyl sulfate (SDS) solution was dripped on the leaf of *N. benthamiana*. The *M. pteropus* plants were separately incubated in water or 10% SDS. The water group was employed as a negative control. The SDS group was employed as a positive control since high SDS concentration could destroy the plant cuticle. *N. benthamiana* and *M. pteropus* were also treated by core-shell MNPs, separately. The plants were grown for 7 days. Leaves were rinsed by DI water before characterization. Chlorophyll concentration was quantified by the atLEAF handheld chlorophyll meter (FT Green LLC, USA) according to a previous report[66]. The treated leaves were collected and dark-adapted for 15 min. OJIP curves were generated by FluorPen FP 110 (Photon System Instruments, Czech Republic). Photosynthetic performance indexes (Fv/Fm, NPQ, qP and ETR) were analyzed using Closed FluorCam 1300 (Photon System Instruments, Czech Republic).

## Statistics and reproducibility

All data are presented as mean ± standard deviation (s.d.) unless otherwise stated. Statistical analyses were performed using OriginPro 2021. Comparisons between two groups were evaluated using two-tailed Student's *t*-test. Comparisons among three or more groups were evaluated using one-way analysis of variance (ANOVA) followed by Tukey's multiple comparisons test. Statistical significance was defined as ns ($P > 0.05$), * ($P < 0.05$), ** ($P < 0.01$), *** ($P < 0.001$), and **** ($P < 0.0001$), with a 95% confidence interval.

## Reporting summary

Further information on research design is available in the Nature Portfolio Reporting Summary linked to this article.

## Data availability

Data supporting the findings of this work are available within the paper and its Supplementary Information files. A reporting summary for this Article is available as a Supplementary Information file. Source data are provided with this paper.

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

## Acknowledgements

This work was supported by the Singapore National Research Foundation (NRF) under the NRF Fellowship (Award No: NRF-NRFF15-2023-0002, T.T.S.L.) and by the Research Center for Sustainable Urban Farming, NUS (Grant No: A-8000149-00-00, T.T.S.L.). Schematic images were partially created using BioRender (https://www.biorender.com/).

## Author contributions

D.S. and T.T.S.L. conceptualized and designed the study. D.S., S.R., Y.Z. and Z.P. performed the experiments. D.S. conducted the computational simulation study of MNP cargo release. S.R. and Y.Z. assisted with plasmid preparation, RT-qPCR and Western blot analysis. Y.L., C.T., S.P. and C.S. assisted with MNP characterization. D.S. and T.T.S.L. wrote the first draft of the paper. All authors contributed to the discussion and revision of the paper.

## Competing interests

The authors declare no competing interests.
