## [Peer Review file · Nature Communications]

Amphibious Microneedles for Programmable Delivery of Biomolecules and Microorganisms in Living Plants

Corresponding Author: Dr Tedrick Thomas Salim Lew

Version 0:

Reviewer comments:

Reviewer #1

(Remarks to the Author)

The authors have provided exceptionally thorough and well-reasoned responses to all reviewer comments. They not only addressed each point in detail but also conducted substantial new experiments that significantly strengthen the manuscript. Overall, the revisions fully resolve my previous concerns, and the manuscript is ready for publication.

Reviewer #2

(Remarks to the Author)

The referee appreciates all the work done to improve the quality of the manuscript, and has no further comment.

Reviewer #3

(Remarks to the Author)

In my initial review, I raised several concerns regarding the novelty of the microneedle platform, the consistency of microneedle function, the use of potentially improper GFP imaging, the lack of appropriate experimental controls, and the practical applications of the MNP system. The authors have now substantially revised the manuscript and conducted a broad set of new experiments and analyses that directly and convincingly address these comments.

For instance, the current version incorporates Scheme S4 and comparative data (hydrogel, hydrophobic, and non-responsive microneedles), which clearly distinguish this platform both mechanistically and functionally from prior plant microneedle systems.

Overall, the authors have fully addressed my earlier concerns with new experiments, stronger contextual positioning, and improved clarity. I feel the manuscript now makes a timely and significant contribution to plant biotechnology, introducing a unique, amphibious microneedle system for programmable delivery under challenging wet and submerged conditions.

A couple of minor points:

1) In Figure 3, the authors used a 35S-GFP vector to demonstrate plasmid delivery via MNPs. Could the authors clarify whether the same GFP construct was used for *N. benthamiana* and *M. Pteropus* in Figure 4? Does this construct include an NLS (nuclear localization signal) tag? If so, the strong nuclear-localized GFP fluorescence observed in *N. benthamiana* is expected. However, it is less clear why *M. pteropus* shows no nuclear GFP fluorescence signal after GFP delivery. GFP is small enough to go into the nucleus even without an NLS. This should be clarified for readers to avoid confusion.

2) The new experiments delivering GFP-tagged AoCLCf construct to aquatic plants under salt stress are trying to provide compelling evidence that the MNP system has real biological applications. However, the interpretation of Figure 5 (Lines 390–413) does not fully align with the figure panels. For instance, MNP-A group shows salt resistance in Figure 5a and high GFP expression levels in Figure 5b, but Figure h6 does not display any detectable GFP fluorescence. This apparent mismatch between quantitative data and representative images should be confirmed and clarified. The authors should carefully check the arrangement and labeling of Figure 5 panels and ensure the accompanying text consistently and accurately interprets the data.

With these clarifications, I believe the manuscript is acceptable for publication.

Version 1:

Reviewer comments:

Reviewer #3

(Remarks to the Author)

I'd like to thank the authors for answering my additional questions. I have no more concerns for this manuscript. It is ready for publication.

Point-by-point Responses

Manuscript title: “Amphibious Microneedles for Programmable Delivery of Biomolecules and Microorganisms in Living Plants”

REVIEWER COMMENTS

Reviewer 1

The authors have provided exceptionally thorough and well-reasoned responses to all reviewer comments. They not only addressed each point in detail but also conducted substantial new experiments that significantly strengthen the manuscript. Overall, the revisions fully resolve my previous concerns, and the manuscript is ready for publication.

[Authors]: We sincerely thank the reviewer for the positive and encouraging feedback.

Reviewer 2

The referee appreciates all the work done to improve the quality of the manuscript, and has no further comment.

[Authors]: We thank the reviewer for the kind acknowledgment of our efforts to improve the manuscript and for the positive assessment.

Reviewer 3

In my initial review, I raised several concerns regarding the novelty of the microneedle platform, the consistency of microneedle function, the use of potentially improper GFP imaging, the lack of appropriate experimental controls, and the practical applications of the MNP system. The authors have now substantially revised the manuscript and conducted a broad set of new experiments and analyses that directly and convincingly address these comments. For instance, the current version incorporates Scheme S4 and comparative data (hydrogel, hydrophobic, and non-responsive microneedles), which clearly distinguish this platform both mechanistically and functionally from prior plant microneedle systems.

Overall, the authors have fully addressed my earlier concerns with new experiments, stronger contextual positioning, and improved clarity. I feel the manuscript now makes a timely and significant contribution to plant biotechnology, introducing a unique, amphibious microneedle system for programmable delivery under challenging wet and submerged conditions.

[Authors]: We sincerely thank the reviewer for the positive and encouraging comments. We greatly appreciate the reviewer’s appreciation of our extensive revisions and new experiments.

Minor Concerns:

1. In Figure 3, the authors used a 35S-GFP vector to demonstrate plasmid delivery via MNPs. Could the authors clarify whether the same GFP construct was used for *N. benthamiana* and *M. Pteropus* in Figure 4? Does this construct include an NLS (nuclear localization signal) tag? If so, the strong nuclear-localized GFP fluorescence observed in *N. benthamiana* is expected. However, it is less clear why *M. pteropus* shows no nuclear GFP fluorescence signal after GFP delivery. GFP is small enough to go into the nucleus even without an NLS. This should be clarified for readers to avoid confusion.

Point-by-point Responses

[Authors]: We appreciate the reviewer's feedback. We confirm that the same GFP construct is used in this work, and it does not include the NLS tag. *N. benthamiana* is a well-established model plant with high 35S promoter activity. The 35S-GFP construct has been widely validated on *N. benthamiana* for gene delivery studies and typically produces strong nuclear-localized GFP fluorescence. In contrast, aquatic *M. pteropus* showed primarily cytoplasmic fluorescence without a clear nuclear signal. This difference likely arises from the distinct cellular architecture and physiological environment of *M. pteropus* under aquatic growth conditions, which may alter nuclear permeability, cytoplasmic viscosity, and GFP folding or maturation efficiency. (*Current Plant Biology*, 2020, 24, 100179; *Plant Physiology*, 2013,163, 648-658; *Molecular Biology Reports*, 2021, 48, 2235-2241). Nevertheless, Western blot revealed a clear GFP band (**Fig. S31**), indicating successful protein expression. RT-qPCR also detected GFP mRNA (**Fig. 4f, g**), supporting effective plasmid delivery and transcription in *M. pteropus*. These results could collectively demonstrate that our microneedle platform enables successful delivery of plasmids and *Agrobacterium* into *M. pteropus*.

We have included this discussion in the revised manuscript:

Lines 309-313: "Interestingly, *M. pteropus*, unlike *N. benthamiana*, exhibited predominantly cytoplasmic fluorescence with minimal nuclear localization under confocal microscopy. This difference likely arises from the distinct cellular architecture and physiological environment of *M. pteropus* under aquatic growth conditions, which may alter nuclear permeability, cytoplasmic viscosity, and GFP folding or maturation efficiency.⁴³⁻⁴⁵."

2. The new experiments delivering GFP-tagged AoCLCf construct to aquatic plants under salt stress are trying to provide compelling evidence that the MNP system has real biological applications. However, the interpretation of Figure 5 (Lines 390–413) does not fully align with the figure panels. For instance, MNP-A group shows salt resistance in Figure 5a and high GFP expression levels in Figure 5b, but Figure h6 does not display any detectable GFP fluorescence. This apparent mismatch between quantitative data and representative images should be confirmed and clarified. The authors should carefully check the arrangement and labeling of Figure 5 panels and ensure the accompanying text consistently and accurately interprets the data.

[Authors]: We thank the reviewer for this point, and would like to clarify that **Fig. 5h** shows reactive oxygen species (ROS) accumulation under salt stress, not GFP fluorescence. In **Fig. 5h**, the plants were stained with non-fluorescent DCFH-DA, which could be oxidized by ROS to generate fluorescent DCF. MNP-P (**Fig. 5h4**) and MNP-A (**Fig. 5h6**) groups show significantly weakened fluorescence than the other treatments, which support the successful expression and stress-mitigating function of the GFP-AoCLCf construct.

The confocal images of GFP-AoCLCf expression are instead presented in **Fig. S38**, where clear GFP fluorescence can be observed in both MNP-P (**Fig. S38d**) and MNP-A (**Fig. S38f**) groups, consistent with the results of **Fig. 5a** and **Fig. 5b**.